# Post-Serpentinization Formation of Theophrastite-Zaratite by Heazlewoodite Desulfurization: An Implication for Shallow Behavior of Sulfur in a Subduction Complex

**Shoji Arai** [1,*] , **Satoko Ishimaru** [2], **Makoto Miura** [3], **Norikatsu Akizawa** [4] **and Tomoyuki Mizukami** [5]

1   Kanazawa University, Kanazawa 920-1192, Japan
2   Department of Earth Sciences, Kumamoto University, Kumamoto 860-8555, Japan; s_ishimaru@kumamoto-u.ac.jp
3   Identification Department, GIA Tokyo Godo Kaisha, Yamaguchi Building 7, 4-19-9 Taito, Taito-ku, Tokyo 110-0016, Japan; makomiu1214@gmail.com
4   Department of Ocean Floor Geoscience, Atmosphere and Ocean Research Institute, The University of Tokyo, 5-1-5 Kashiwanoha, Kashiwa, Chiba 277-8564, Japan; akizawa@aori.u-tokyo.ac.jp
5   Department of Earth Sciences, Kanazawa University, Kanazawa 920-1192, Japan; peridot@staff.kanazawa-u.ac.jp
*   Correspondence: ultrasa@staff.kanazawa-u.ac.jp

**Abstract:**     Rare nickel hydroxide-hydroxyl carbonate, theophrastite ($Ni(OH)_2$)-zaratite ($Ni_3(CO_3)(OH)_4\cdot4H_2O$) aggregates were found from a partially serpentinized dunite from Fujiwara, the Sanbagawa metamorphic belt of high-pressure intermediate type, Japan. The dunite was regionally metamorphosed within the Sanbagawa subduction complex of Cretaceous age. The theophrastite-zaratite aggregate from Fujiwara most typically occurs in association with nickel sulfides, which form a composite grain with awaruite and magnetite within an antigorite-rich part of the rock. The theophraste-zaratite formed possibly together with millerite (NiS) from heazlewoodite ($Ni_3S_2$). This represents a partial desulfurization of heazlewoodite, which contains or interlocks with laths of antigorite, suggesting their cogenesis. The desulfurization occurred at an early stage of, or during, exhumation of the subduction complex toward the surface, where sulfur was oxidized and removed as sulfate ions. Serpentinization of olivine has not been associated with the formation of theophrastite-zaratite, and an oxidized condition has been kept at this post-serpentinization stage. The sulfate ions liberated in part precipitated anhydrite where calcium was available in the surrounding rocks. This shows one of the shallow migration pathways of sulfur in the subduction zone, especially to the forearc area.

**Keywords:**  theophrastite; zaratite; serpentinization; heazlewoodite; millerite; desulfurization; subduction; exhumation; Sanbagawa metamorphic belt; Japan

---

## 1. Introduction

The behavior of sulfur in a subduction zone is very important and has been controversial [1–3]. Sulfur is transported to deeper parts via the subduction of sediments and altered crustal rocks such as meta-basalts [4]. It is also taken from sea water into peridotite on sea-floor serpentinization [5–7], and serpentinized peridotites gradually release sulfur upon progressive metamorphism in the subducted slab [8]. This is consistent with the relative sulfur enrichment in arc magmas [9]. However,

the manners of delivery of sulfur from the slab to the overlying crust-mantle have not been thoroughly delineated so far. We would like to show one of the ways of sulfur recycling through heazlewoodite, a nickel sulfide, in subducted serpentinite, possibly to the supra-subduction zone.

Nickel is compatible together with magnesium in a magmatic system and concentrated in early formed mafic minerals, especially in olivine [10] in addition to some sulfides. On the low temperature hydration of Mg-rich olivines, nickel then released from olivine is reduced to be crystallized mainly as Ni-Fe alloys (awaruites) and/or sulfides such as heazlewoodite [11–14]. Hydrogen is simultaneously produced, coupled with magnetite precipitation in the serpentinization process [15].

Theophrastite is a rare nickel hydroxide, first described from highly altered chromitite (magnetite-chromite ore) from northern Greece [16,17], as well as from altered chromitites in Unst, Shetland (Scotland) [18]. In addition to theophrastite, hydroxyl nickel carbonates such as zaratite [$Ni_3CO_3(OH)_4 \cdot 4H_2O$], otwayite [$Ni_2CO_3(OH)_2 \cdot H_2O$], nullaginite [$Ni_2CO_3(OH)_2$] and hellyerite [$NiCO_3 \cdot 6H_2O$] and Ni-rich carbonates such as gaspéite [$(Ni, Fe, Mg)CO_3$] are precipitated during low-temperature alteration or weathering of serpentinites and related nickel-rich ores [19–24]. In low-temperature alteration processes of serpentinites, nickel (II) is also incorporated in hydrous silicates (clay minerals) such as pecoraite [$Ni_3SiO_5(OH)_4$] and nickel-rich sepiolite around nickel sulfides/oxides [21–25].

All of the previous works, however, have given us rather weak genetic constraints on theophrastite and hydroxyl nickel carbonates, as well as weak petrological implications. Aggregates of theophrastite, $Ni(OH)_2$, and a nickel hydroxyl carbonate (zaratite = $Ni_3(CO_3)(OH)_4 \cdot 4H_2O$) were found in a partly serpentinized (antigoritized) dunite from Fujiwara (Ehime Prefecture), the Sanbagawa metamorphic belt, southwest Japan (Figure 1). They show a very clear mode of occurrence, that is, a close association with nickel sulfides (heazlewoodite and millerite). The Fujiwara theophrastite-zaratite aggregate seems quite different in the mode of occurrence from the other reported occurrences, which were mostly along shear planes in altered chromitites and nickel ores. In this article, we describe the nickel-rich minerals, especially theophrastite-zaratite (nickel hydroxide-hydroxyl carbonate), in a partly antigoritized dunite, and discuss their origin with an implication of their occurrence in a subduction complex. More general petrological characteristics of the Fujiwara meta-dunites are available from Arai et al. [26].

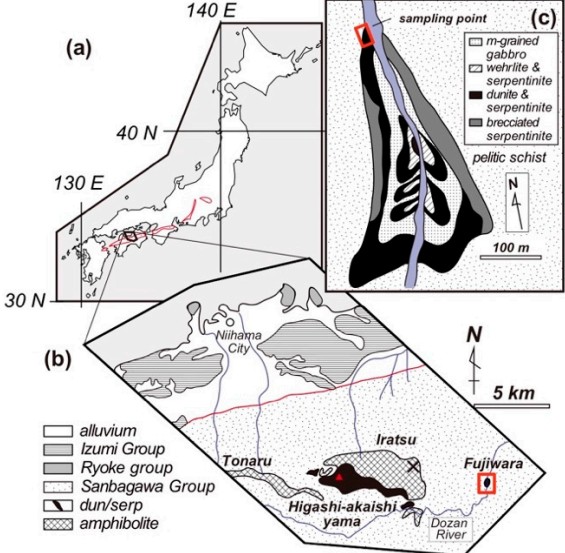

**Figure 1.** Location of the Fujiwara dunite in the Sanbagawa metamorphic belt, Japan. Modified from [27,28]. (**a**) The Sanbagawa metamorphic belt in Shikoku Island, southwest Japan. (**b**) Part of the Sanbagawa belt and the location of the Fujiwara body (**c**).

## 2. Geological Background and Petrography

The dunite sample (T4-8) was obtained from the Fujiwara body, one of the abundant peridotite masses [29,30] in the Sanbagawa metamorphic belt in Shikoku Island, southwest Japan [31,32] (Figure 1). They are associated with crystalline schists of high-pressure intermediate type [29], representative of a subducting slab complex of the Cretaceous age. The Fujiwara dunite body (less than 200 × 400 m in plan) is associated with fine- to coarse-grained (mainly medium-grained) gabbros to form as a whole an ultramafic-mafic complex [26], enclosed by pelitic schists of a high-temperature part of the biotite zone to garnet-biotite transition zone [33]. The Fujiwara dunite suffered from the Sanbagawa high-pressure/low-temperature metamorphism [31,32], together with the surrounding schists [33], like other peridotites bodies in the Sanbagawa belt [29].

Detailed descriptions were made on one of the dunite samples (meta-dunite; T4-8) containing the theophrastite-zaratite aggregate (Figure 2; Figure 3), which is fresh and composed of olivine, chromian spinel (altered in part to ferritchromite and magnetite), titanoclinohumite, antigorite, magnetite, chlorite and dolomite [26,28,33]. The examined sample piece is completely free from a weathered (brownish-colored) portion. Serpentinization has proceeded in an uneven way; the rock consists of antigorite-rich and fresh olivine-rich domains. Antigorite occurs as laths partly penetrating adjacent olivine grains (Figure 2), if any, and occupies around 40 volume % of the rock on average. Awaruite and nickel sulfides (heazlewoodite and millerite) were found, associated with theophrastite-zaratite (Figure 4), as described in detail later. Minute rounded particles of nickel arsenide minerals were frequently included by heazlewoodite and magnetite (Figure 5).

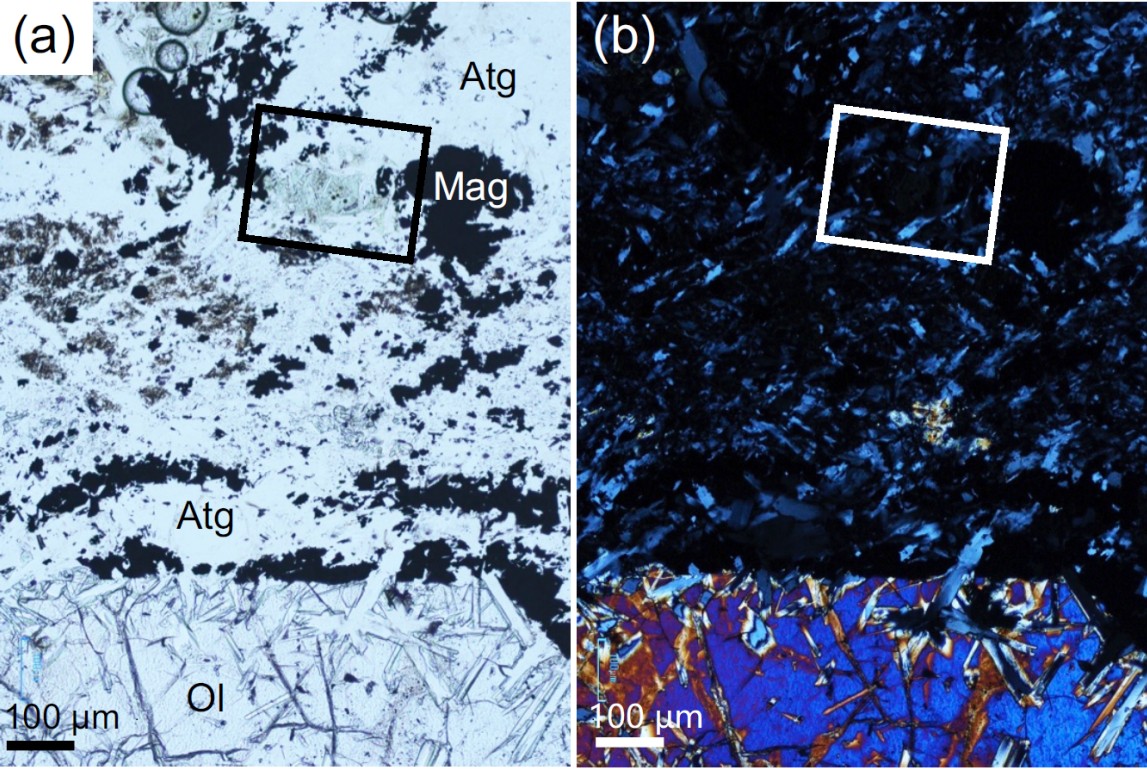

**Figure 2.** Photomicrographs of the partially serpentinized dunite that contains theophrastite-zaratite aggregates from Fujiwara, the Sanbagawa metamorphic belt, southwest Japan. Atg, antigorite. Mag, magnetite. Ol, olivine. (**a**) Antigorite-rich part with relic olivine. Note the penetration of antigorite laths into olivine. Plane-polarized light. The theophrastite-zaratite aggregate was found in the rectangle, which is enlarged as Figure 3. (**b**) Crossed-polarized light image of (**a**).

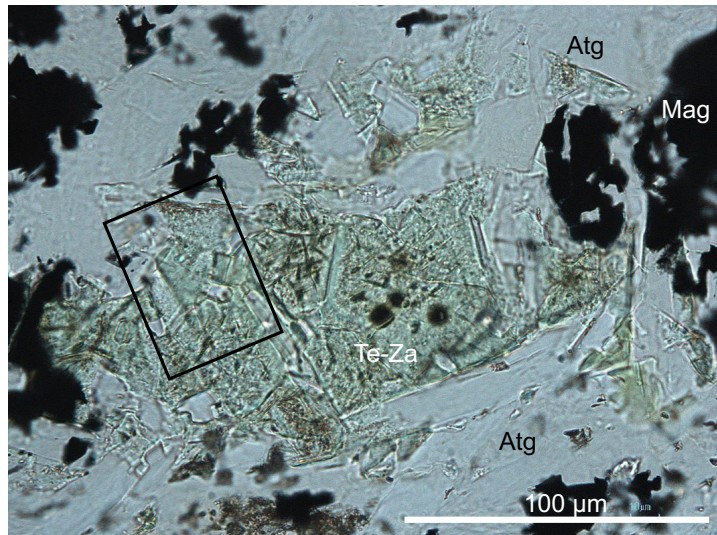

**Figure 3.** Discrete grain of theophrastite-zaratite aggregate (light green; Te-Za) within an antigorite-rich part. Note the penetration of antigorite laths into the aggregate. Dark spots represent damages due to the evaporation of volatiles in microprobe analysis. Plane-polarized light. The rectangle shows an area for Raman spectroscopic analysis below.

The nickel hydroxide-hydroxyl carbonate (theophrastite-zaratite) aggregate is closely associated with heazlewoodite in the opaque composite grains, composed of awaruite, millerite and magnetite, in addition to heazlewoodite (Figure 4). The heazlewoodite is mottled with veinlet-like or irregular-shaped patches of millerite (Figures 4 and 5), indicating that the millerite in part replaces heazlewoodite. Awaruite, millerite and magnetite form an apparent rim of heazlewoodite and/or theophrastite-zaratite (Figure 4). The theophrastite-zaratite aggregate is characterized by prominent cracking (Figures 4 and 5). This cracking has been frequently observed in PGE oxides produced by desulfurization of PGE sulfides, such as laurite [34–36].

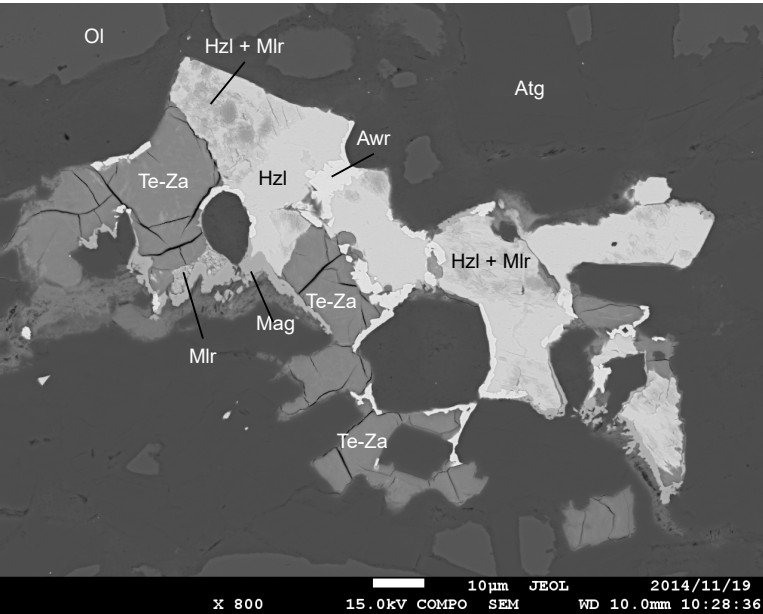

**Figure 4.** SEM (scanning electron microscope) image of a composite grain containing theophrastite-zaratite aggregate (Te-Za). Note the cracking in the Te-Za aggregate. Awr, awaruite. Hzl, heazlewoodite. Mlr, millerite. Mag, magnetite. Ol, olivine. Atg, antigorite. Bar is 10 μm.

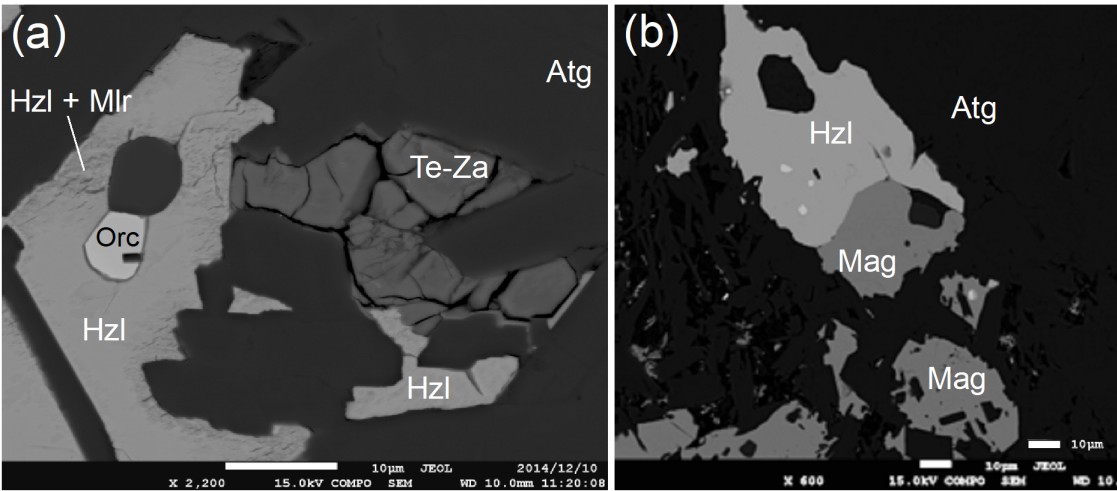

**Figure 5.** SEM images of heazlewoodite and related minerals. (**a**) Theophrastite-zaratite aggregate (Te-Za) associated with heazlewoodite. Note the inclusion of orcelite (Orc; arsenide) which contains am antigorite lath. Hzl, heazlewoodite. Mlr, millerite. Atg, antigorite. (**b**) Magnetite (Mag) associated with heazlewoodite. Note that the antigorite laths included by Hzl and Mag. Bright dots in Hzl are orcelite.

We found discrete grains of theophrastite-zaratite aggregate, which are appropriate to examine optical properties (Figures 2 and 3). The largest is 0.1 to 0.2 mm across, and shows a pale green color, no pleochroism and very low birefringence (almost isotropic) in thin section (Figures 2 and 3). It is similar in appearance to a kind of chlorite under the microscope (Figures 2 and 3). It occurs in one of the antigorite-rich domains, mainly composed of antigorite and magnetite, of the rock (Figure 2). The aggregate encloses minute laths of antigorite (Figure 3). The optical properties (especially refractive indices, birefringence, and colors) of the aggregate are similar to those of zaratite [20] or the Greek theophrastite [16].

The mode of occurrence of theophrastite-zaratite in the Fujiwara meta-dunite is quite different from the other theophrastite occurrences available from the literature. The theophrastite from Greece exhibits a botryoidal texture, sometimes forming fibrous aggregates perpendicularly overgrowing on pre-existing minerals such as vesuvianite and chlorite in magnetite-chromite ores (= altered or metamorphosed chromitite?) [16,17]. Theophrastites from Shetland [18] and from Heazlewood River [19] form gel (or wax)-like mixtures with other low-temperature (sometimes poorly crystalline) nickel hydroxyl carbonates, such as zaratite or otwayite in altered chromitites.

## 3. Raman Spectroscopy

We conducted a Raman spectroscopic analysis on the discrete grain of nickel hydroxide-hydroxyl carbonate. We used a micro-Raman system (HORIBA Jobin Yvon, LabRAM HR800) equipped with an optical microscope (Olympus, BX41) at Kanazawa University, Kanazawa, Japan. An 830 nm semiconductor laser (Sacher Lasertechnik, TEC510) and an IR detector (512 × 1 pixels of InGaAs elements) were used, in order to obtain Raman spectra without the interference of a photoluminescence band of nickel hydroxide or carbonate (maximized at 593 nm). The laser beam was focused through a 100× objective lens for infrared (LEICA GERMANT, HCX PL FLUOTAR, 100×, NA = 0.75), yielding an irradiation power of 2.0 mW at the sample surface. Scattered light was collected in a backscattered geometry, in which a pinhole (300 μm in diameter) and a slit (100 μm in width) were positioned in front of the spectrometer. A 300 grooves/mm grating with a blaze wavelength of 1000nm gave a wavenumber resolution of 1.8–2.3 cm$^{-1}$ and a spectral resolution of about 3.3 cm$^{-1}$ to 5.0 cm$^{-1}$. An edge filter effectively reduces the Rayleigh and anti-Stokes lines to lower than 70 cm$^{-1}$ in Raman shift. For mapping analysis, a 514.5 nm Ar$^{+}$ laser (MELLES GRIOT, 43 SERIES ION LASER, 543-GS-A02) and a Si-based CCD (charge-coupled device) detector (1024 × 256 pixels) were used. A combination of a 100× objective lens (Olympus, MPLN100×, NA = 0.90 for visible light) and a 300-μm pinhole

yields a spatial resolution of about 1 μm. The laser irradiation power at a sample surface was 9.2 mW. The spectrometer was calibrated using a Si 520 cm$^{-1}$ peak before measurements. This procedure provided a sufficient accuracy in the analytical wavenumber range; the peak centers of neon lines were reproduced within ±0.5 cm$^{-1}$.

The Fujiwara theophrastite-zaratite and associated minerals were examined by Raman spectroscopy, although there have been few data on theophrastite for comparison [37,38]. As far as we know, no Raman spectroscopic characteristics of natural theophrastite have ever been published. We obtained Raman spectra of theophrastite from a part of the aggregate (Figure 6). They are characterized by a prominent band at 3580 cm$^{-1}$ caused by OH stretching vibrations [37] (Figure 6), and some minor bands due to low-energy translational modes were observed at <1600 cm$^{-1}$ (Figure 6). Almost all Raman spectral bands of the Fujiwara theophrastite are in agreement with those of synthetic $Ni(OH)_2$ [37,38]. The other part of the aggregate yielded strong photoluminescence for some reason, but showed some bands derived from zaratite [39] and unknown mineral(s) (Figure 6). No clear spectra for other hydroxyl nickel carbonates, such as nullangite [$Ni_2(CO_3)(OH)_2$] and otwayite, were detected by Raman spectroscopy [39,40]. This is consistent with the optical properties of the aggregate (Figure 3) similar to those of zaratite [20,21].

Our Raman spectroscopic analysis confirmed the serpentine mineral associated with the theophrastite-zaratite is antigorite [41] in the Fujiwara meta-dunite (Figures 2 and 3). Olivine and titanoclinohumite grains contain micro-inclusions of hydrocarbons (methane and propane), serpentine and brucite [28]. They were originally trapped as hydrocarbon-bearing aqueous solutions in olivine and titanoclinohumite [28].

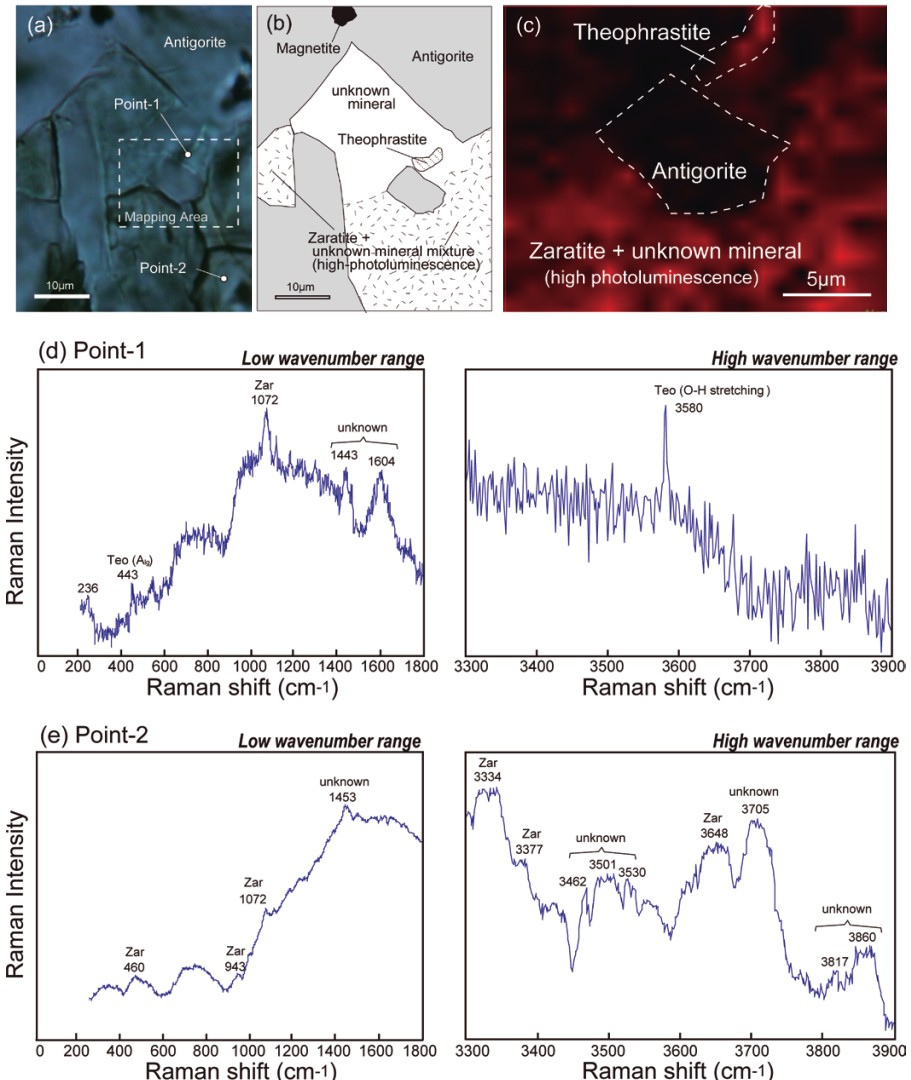

**Figure 6.** Laser Raman spectroscopy of the theophrastite-zaratite aggregate in the dunite from Fujiwara, Japan. (**a**) Plane-polarized image of the examined area, which is equivalent to the rectangle in Figure 3. The rectangle shows the area for Raman spectroscopic mapping (**c**). Spectra for points 1 and 2 are shown in panels (**d**,**e**), respectively. (**b**) Summary of identification of mineral species for the area of panel (**a**). (**c**) Distribution of Raman spectral intensity in terms of wavenumber = 3580 cm$^{-1}$, where one of the main peaks for theophrastite is available [37]. Theophrastite was identified only from an encircled area, but the other area shows high photoluminescence, where only zaratite was identified [39]. (**d**) Raman spectra from Point 1 (panel (**a**)), where theophrastite was identified. Note a prominent band at 3580 cm$^{-1}$ caused by O-H stretching vibrations of theophrastite [37]. (**e**) Raman spectra from Point 2 (panel (**a**)), where only zaratite was identified with high photoluminescence background. Teo, theophrastite. Zar, zaratite.

## 4. Mineral Chemistry

Minerals were mainly analyzed by a JEOL wave-length dispersive electron probe X-ray microanalyzer (JXA8800R) at Kanazawa University, Kanazawa, Japan. Analytical conditions were 15-kV accelerating voltage, 10-nA accelerating voltage, and 10-μm probe diameter for analysis of the nickel hydroxide-hydroxyl carbonate to minimize evaporation loss during analysis. A 5-μm probe diameter was adopted for antigorite analysis. 12-nA probe current and 3-μm probe diameter were used for other minerals.

Ferrous and ferric iron contents of chromian spinel were calculated assuming spinel stoichiometry, while all iron was assumed to be ferrous in minerals except spinel phases. Mg# is $Mg/(Mg + Fe^{2+})$ atomic ratio, and Cr# is $Cr/(Cr + Al)$ atomic ratio. $Fe^{2+}$ and $Fe^{3+}$ in spinels were calculated assuming spinel stoichiometry. Representative microprobe analyses of minerals are listed in Table 1.

**Table 1.** Selected microprobe analyses of minerals in the meta-dunite (T4-8) from Fujiwara, the Sanbagawa metamorphic belt, Southwest Japan.

| Phase | Ol | | Ti-Chu | Chl | Atg | Chr | Fch | Mag | Te-Za | | |
|---|---|---|---|---|---|---|---|---|---|---|---|
| $SiO_2$ | 40.47 | 40.90 | 37.02 | 34.75 | 44.09 | nd | 0.44 | 0.01 | 0.17 | 0.22 | 0.20 |
| $TiO_2$ | 0.03 | 0.40 | 3.31 | nd | 0.03 | 0.91 | 0.73 | 016 | nd | nd | 0.03 |
| $Al_2O_3$ | nd | nd | nd | 11.76 | 0.40 | 17.51 | 3.16 | 0.03 | 0.01 | nd | 0.01 |
| $Cr_2O_3$ | nd | nd | 0.02 | 0.51 | nd | 35.39 | 31.11 | 0.18 | nd | nd | nd |
| FeO* | 10.95 | 4.95 | 6.63 | 3.20 | 1.16 | 35.66 | 53.49 | 91.81 | 0.15 | 0.28 | 0.81 |
| MnO | 0.79 | 0.21 | 0.35 | 0.01 | 0.04 | 1.18 | 2.50 | 0.17 | nd | nd | nd |
| MgO | 48.36 | 53.14 | 50.78 | 35.52 | 40.35 | 8.33 | 3.73 | 0.98 | 4,17 | 5.05 | 4.49 |
| CaO | 0.01 | 0.01 | 0.01 | 0.01 | nd | 0.02 | nd | nd | 0.37 | 0.31 | 0.26 |
| $Na_2O$ | nd | nd | 0.01 | 0.01 | nd | 0.02 | 0.05 | 0.04 | 0.03 | 0.03 | 0.05 |
| $K_2O$ | nd | nd | nd | 0.01 | 0.01 | nd | nd | nd | nd | nd | nd |
| NiO | 0.41 | 0.27 | 0.13 | 0.10 | 0.11 | 0.22 | 0.53 | 0.66 | 60.45 | 57.22 | 58.97 |
| Total | 101.04 | 99.88 | 98.25 | 85.87 | 86.18 | 99.24 | 95.72 | 94.02 | 65.38 | 63.11 | 64.81 |
| O | 4 | 4 | 13 | 14 | 7 | 4 | 4 | 4 | 10 | 10 | 10 |
| Si | 0.992 | 0.985 | 2.996 | 3.319 | 2.056 | 0.000 | 0.018 | 0.000 | 0.030 | 0.041 | 0.036 |
| Ti | 0.001 | 0.007 | 0.201 | 0.000 | 0.001 | 0.023 | 0.022 | 0.006 | 0.000 | 0.000 | 0.003 |
| Al | 0.000 | 0.000 | 0.000 | 1.323 | 0.022 | 0.701 | 0.153 | 0.002 | 0.001 | 0.000 | 0.003 |
| Cr | 0.000 | 0.000 | 0.001 | 0.038 | 0.000 | 0.950 | 1.011 | 0.007 | 0.000 | 0.000 | 0.000 |
| Fe* | 0.224 | 0.100 | 0.448 | 0.255 | 0.045 | 1.013 | 1.839 | 3.865 | 0.023 | 0.042 | 0.122 |
| Mn | 0.016 | 0.004 | 0.024 | 0.000 | 0.002 | 0.034 | 0.087 | 0.007 | 0.000 | 0.000 | 0.000 |
| Mg | 1.766 | 1.905 | 6.122 | 5.054 | 2.802 | 0.422 | 0.229 | 3.865 | 1.115 | 1.378 | 1.024 |
| Ca | 0.000 | 0.000 | 0.001 | 0.001 | 0.000 | 0.001 | 0.000 | 0.007 | 0.071 | 0.062 | 0.049 |
| Na | 0.000 | 0.000 | 0.001 | 0.002 | 0.000 | 0.002 | 0.004 | 0.000 | 0.010 | 0.009 | 0.018 |
| K | 0.000 | 0.000 | 0.000 | 0.001 | 0.000 | 0.000 | 0.000 | 0.004 | 0.000 | 0.000 | 0.000 |
| Ni | 0.008 | 0.005 | 0.009 | 0.008 | 0.004 | 0.006 | 0.018 | 0.000 | 8.721 | 8.432 | 8.533 |
| Total | 3.008 | 3.008 | 9.803 | 10.002 | 4.932 | 3.152 | 3.338 | 3.991 | 9.973 | 9.964 | 9.968 |
| Mg# | 0.887 | 0.950 | 0.932 | 0.952 | 0.984 | 0.405 | 0.211 | 0.056 | 0.980 | 0.970 | 0.908 |
| Cr# | | | | | | 0.576 | 0.869 | 0.812 | | | |
| $Y_{Cr}$ | | | | | | 0.465 | 0.471 | 0.003 | | | |
| $Y_{Al}$ | | | | | | 0.343 | 0.071 | 0.001 | | | |
| $Y_{Fe}$ | | | | | | 0.192 | 0.458 | 0.997 | | | |

In wt.% for oxides. Ol, olivine. Ti-Chu, titanoclinohumite. Chl, chlorite. Atg, antigorite. Chr, chromian spinel. Fch, ferritchromite. Mag, magnetite. Te-Za, theophrastite-zaratite agggregate. FeO* & Fe*, total iron as FeO & Fe*, respectively. nd, not detected. Mg#, $Mg/(Mg + Fe^{2+})$ atomic ratio. Cr#, $Cr/(Cr + Al)$ atomic ratio. $Y_{Cr}$, $Y_{Al}$ & $Y_{Fe}$, atomic fractions of Cr, Al & $Fe^{3+}$ to $(Cr + Al + Fe^{3+})$.

Sulfides, arsenides and alloys were analyzed with FE-SEM (JSM-7001F, JEOL) with an EDS system (INCA Energy and/or AZtecEnergy, Oxford Instruments) at Kumamoto University, Kumamoto, Japan. The accelerating voltage, probe current, and probe diameter were 15 kV, 1 nA, and <1 μm, respectively. Representative analyses are listed in Table 2.

**Table 2.** Selected SEM analyses of sulfides, arsenides and alloy in the meta-dunite (T4-8) from Fujiwara in the Sanbagawa metamorphic belt, southwest Japan.

| Minerals | Hzl | Mlr | Hzl* | Mlr* | Orc | Mau | Awr |
|---|---|---|---|---|---|---|---|
| Fe | | | 1.66 | 0.66 | 0.31 | | 22.01 |
| Co | 0.69 | | 0.64 | | | | 1.48 |
| Ni | 71.45 | 64.27 | 71.46 | 63.49 | 61.90 | 52.16 | 33.02 |
| Sb | | | | | 3.56 | 3.16 | |
| S | 27.22 | 33.02 | 26.13 | 35.00 | | | 0.15 |
| As | | | | | 34.48 | 45.99 | |
| Total | 99.12 | 97.29 | 99.89 | 99.15 | 100.25 | 101.31 | 97.29 |
| Fe | | | 0.014 | 0.005 | 0.004 | | 0.235 |
| Co | 0.004 | | 0.005 | | | | 0.015 |
| Ni | 0.587 | 0.515 | 0.587 | 0.495 | 0.681 | 0.581 | 0.747 |
| Sb | | | | | 0.019 | 0.017 | |
| S | 0.409 | 0.485 | 0.393 | 0.499 | | | 0.003 |
| As | | | | | 0.297 | 0.402 | |
| Total | 1.000 | 1.000 | 1.000 | 1.000 | 1.000 | 1.000 | 1.000 |
| Ni/(Ni + Fe) | 1.00 | 1.00 | 0.98 | 0.99 | 0.99 | 1.00 | 0.76 |
| Ni/As | | | | | 2.29 | 1.45 | |

Hzl, heazlewoodite; Mlr, millerite; Orc, orcelite; Mau, maucherite; Awr, awaruite. Hzl*, Mlr*, bright and dark parts, respectively, in (Hzl + Mlr) (Figures 4 and 5). Blank, not detected. wt.% and number of atoms (total = 1) in the upper and lower rows, respectively.

The part from which theophrastite-zaratite spectra were obtained by Raman spectroscopy was too small for microprobe analysis (Figure 6). The elemental distributions determined by microprobe (Figure 7) indicate that the zaratite-theophrastite aggregate is homogeneous in C, O, Mg and Ni contents; there is no clear area of $Ni(OH)_2$ (theophrastite) composition (Figure 7). Microprobe analysis of the theophrastite-zaratite aggragate yielded low oxide totals, 63 to 65 wt.%, indicating the presence of substantial amounts of volatiles (Table 1). This is consistent with severe damage due to electron bombardment on polished surface of the theophrastite-bearing part in microprobe analysis (Figure 3). The analytical total is far lower than the ideal value, i.e., about 80.6 wt.% NiO for theophrastite, $Ni(OH)_2$, but is not so different from an ideal value, i.e., about 60 wt.% of NiO for zaratite, $Ni_3(CO_3)(OH)_4 \cdot 4H_2O$. Our ordinary microprobe point analysis also indicates chemical homogeneity of the theophrastite-zaratite aggregate (Figure 7). The aggregate contains 57 to 60 wt.% NiO and 4 to 5 wt.% MgO (Table 1). FeO (total iron) and CaO contents are low, up to 0.8 wt.% and up to 0.4 wt.%, respectively (Table 1). Sulfur was not detected from the Fujiwara theophrastite-zaratite.

Combined with the result of Raman spectroscopy and optical characters, the theophrastite-bearing aggregate is mostly composed of hydroxyl nickel carbonate. Theophrastite is frequently associated with nickel carbonates in previous descriptions. The Shetland theophrastite also occurs as a mixture with zaratite [18]. The Heazlewood (Tasmania) theophrastite is also finely mixed with otwayite, and difficult to analyze separately [19]. The theophrastite from Greece is rather exceptional, showing almost pure $Ni(OH)_2$ in chemistry [16,17].

Hydrous silicates in the dunite are high-Mg#, 0.98 for antigorite and 0.95 for chlorite (Table 1). Both antigorite and chlorite are poor in nickel, containing around 0.1 to 0.2 wt.% NiO (Table 1). Olivine highly varies in Fo content (100Mg#), from 89 to 95, negatively correlated with NiO (0.4 to 0.2 wt.%) and MnO (0.8 to 0.2 wt.%). Titanoclinohumite is high in Mg# (0.93 to 0.96), and contains up to 4 wt.% $TiO_2$ [28]. Chromian spinel exhibits around 0.4 Mg#, 0.5 to 0.6 Cr#, and is characterized by high $TiO_2$ contents, 1 to 3 wt.%. The NiO content is 0.2 to 0.3 wt.% in chromian spinel, around 0.5 wt.% in ferritchromite, and 0.5 to 2 wt.% in magnetites (Table 1).

Awaruite shows the Ni/(Ni + Fe) ratio (in atom) of 0.76, and contains 1 to 2 wt.% Co (Table 2). One of the arsenides shows a Ni/As ratio from 2.3 to 2.6, and is possibly orcelite ($Ni_{5-x}As_2$). It contains about 0.3 wt.% Fe (Table 2). The other exhibits a Ni/As ratio around 1.4 and is possibly maucherite

($Ni_{11}S_8$). Both the orcelite and maucherite contain 2 to 3 wt.% Sb (Table 2). They are similar in chemistry to those described in the literature [42–46]. Kadota et al. [46] especially reported both orcelite and maucherite from the Higashi-akaishi peridotite in the Sanbagawa belt, Shikoku island, located less than 10 km to the west of the Fujiwara peridotite body (Figure 1).

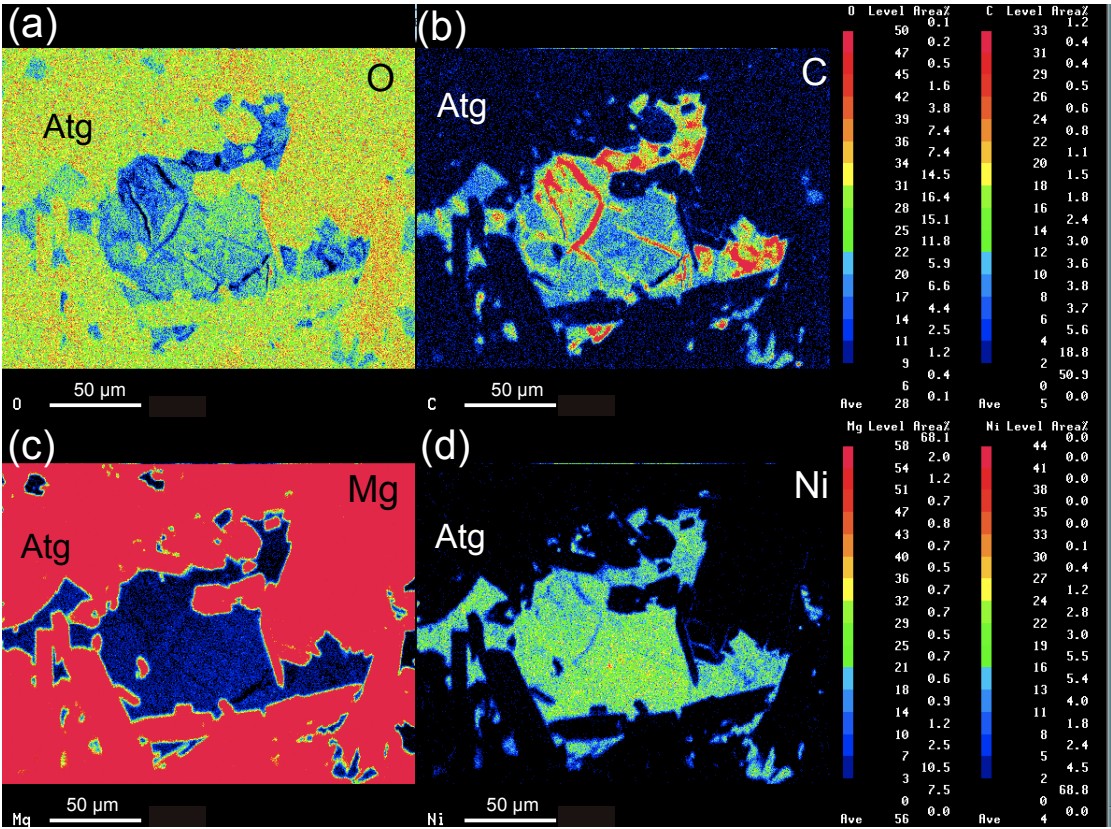

**Figure 7.** Distributions of elements (O, C, Mg and Ni) determined by microprobe in the zaratite-theophrastite aggregate and surroundings. The beam size was less than 1 (nominally 0) μm across, and the scanning step was 1 μm in microprobe analysis. The analyzed field is almost equivalent to that of Figure 3. Warmer colors represent higher contents than cooler colors. (**a**) O. Note that the low O content along fine cracks in the zaratite-theophrastite aggregate was due to remnants of carbon coating for previous microprobe analysis. (**b**) C. Note the sporadic high C contents (red to orange colors) due to incomplete cleaning of carbon coating for previous microprobe analysis. (**c**) Mg. Note the low and homogeneous Mg content in the aggregate. (**d**) Ni. Note the homogeneous Ni content in the aggregate.

## 5. Discussion

### 5.1. Behavior of Nickel during Serpentinization of Olivine

Serpentinization of peridotites usually produces reduced Ni-Fe minerals such as awaruites ($FeNi_3$), and/or sulfides if sulfur is available [12,14,47,48]. Arsenides such as nickeline, orcelite and maucherite, are widely produced in addition, although very small in amount if arsenic is available [42,45]. Arsenic and antimony can be supplied via fluids to peridotite during serpentinization in a subduction zone [49]. Awaruites may be stable in a wide range of serpentinization process, being associated with both antigorite at relatively high temperatures and lizardite at lower temperatures [12,50]. The source of nickel and iron in those minerals is mostly olivine [12,51–53]. Magnetite is one of the products of serpentinization of peridotites [14,54,55]. In our Fujiwara meta-dunite, nickel (II) was also incorporated in magnetite, which contains 0.5 to 0.8 wt.% NiO (Table 1). The ferritchromite ($Fe^{2+}_2CrFe^{3+}O_4$), an alteration product of primary chromian spinel during serpentinization [56,57], is also slightly

enriched with NiO (0.5 wt.%) as compared with the precursor primary chromian spinel, containing 0.2 to 0.3 wt.% NiO (Table 1). The ultimate source of nickel in the Fujiwara theophrastite was olivine, because it is closely associated with antigorite (Figures 3 and 4), which is lower in NiO (0.1 to 0.2 wt.%) than olivine (0.2 to 0.4 wt.%).

In the Sanbagawa subduction complex, subducted sulfur and arsenic have been fixed, at least in part, as nickel-rich sulfides and arsenides during serpentinization of peridotite in the subduction zone. The sulfur and arsenic were originally contained in sediments, altered basalts or ocean-floor serpentinites [4–6], and then incorporated in peridotites on dehydration of the surrounding rocks.

## 5.2. Origin of Theophrastite-Zaratite in the Fujiwara Meta-Dunite

The genetic relationship between theophrastite (nickel hydroxide) and zaratite (one of nickel hydroxyl carbonates) is not clear. Zaratite is possibly a lower temperature alteration product of theophrastite [cf. [21]], but, alternatively, it was co-precipitated with theophrastite with an involvement of $H_2O$-$CO_2$ solution. Stability and formation have been less constrained for nickel carbonates or hydroxyl carbonates than for theophrastite (cf. [22,23,58,59]). We will discuss the origin of theophrastite-zaratite and its implication for mobility of elements, especially sulfur, mainly after antigorite formation (serpentinization).

The textural characteristics suggest the coexistence of antigorite with heazlewoodite, magnetite and arsenides, because the antigorite laths are in part or wholly enclosed by the other minerals (Figures 4 and 5). Awaruites may be cogenetic with them during serpentinization [12,48], although apparently rimming the composite grains in our sample (Figure 4). The millerite, mottling the interior of heazlewoodite, may be secondary to the heazlewoodite (Figures 4 and 5). Antigorite seems to be in textural equilibrium with the discrete grain of theophrastite-zaratite (Figure 3), which is possibly pseudomorphous after heazlewoodite.

Theophrastite is possibly stable only at low temperatures (<247 °C at 1 atm) [60], but can coexist in equilibrium with antigorite, whose low-temperature stability limit is around 200 °C [61]. The Greek theophrastite was estimated to have formed at 80 to 115 °C [17]. Economou-Eliopoulos [62] also suggested an oxidizing condition for the low-temperature magnetite-chromite ore (or Fe-Ni laterites) from northern Greece, from which theophrastite was first described [16]. Nickel-rich alloys are absent, and chlorites, serpentines and magnetites instead contain appreciable amounts of nickel in the Greek Fe-Ni laterites [62]. Unfortunately, serpentine species have not been distinguished, but the surrounding rocks are crystalline schists of greenschist facies [62]. Their protolith was possibly antigorite-bearing meta-peridotite, more or less similar to the Fujiwara dunite.

The formation of theophrastite and anhydrous or hydroxyl nickel carbonates has been attributed to low-temperature alteration, because they formed along fissure or shear planes of serpentinite and related chromitite or nickel ores [19,20,22,23,59,60,63]. They sometimes fill fissure or shear planes or coat metamorphic minerals as weakly crystalline materials near the shear [22,58]. Some of them are not indigenous, and nickel-bearing solutions have been considered to be involved in their formation [18,63]. In contrast, the Fujiwara theophrastite-zaratite in situ formed, replacing the precursor heazlewoodite. The theophrastite from Unst, Shetland, is associated with heazlewoodite, which possibly supplied nickel in its formation [18].

## 5.3. Desulfurization of Heazlewoodite and the Behavior of Sulfur

Petrographic features suggest that desulfurization of heazlewoodite formed theophrastite-zaratite and millerite (Figure 8), that is, via a reaction such as

$$Ni_3S_2 \text{ (heazlewoodite)} + 2H_2O + 2O_2 = NiS \text{ (millerite)} + 2Ni(OH)_2 + SO_2.$$

If $H_2O$-$CO_2$ is involved, zaratite will be produced in addition to theophrastite.

This reaction means a stepwise desulfurization of a nickel sulfide, i.e., the formation of a less sulfur-rich nickel sulfide and theophrastite at the first step (Figure 8). This type of desulfurization of heazlewoodite was reproduced by experiments [64]. Awaruite, which is sometimes associated with theophrastite-zaratite (Figure 4), was possibly not involved in the formation of theophrastite-zaratite via desulfurization of heazlewoodite, which is very low in Fe content (Table 2). We expect that this type of reaction possibly occurs more widely. The partial replacement of heazlewoodite with millerite was also observed in an altered serpentinite that contain hydroxyl nickel carbonates (zaratite and hellyerite) in an altered serpentinite from Heazlewood, Tasmania [20]. Nickel et al. [23] observed the partial replacement of millerite (NiS) with polydymite ($Ni_3S_4$) in a nickel-mineralized part of a serpentinized peridotite, Western Australia, where hydroxyl nickel carbonates (nullangite and otwayite) and related nickel-rich minerals were found. An idealized reaction for these processes can be shown as

Ni-rich sulfide + $H_2O$ + $CO_2$ + $O_2$ = less Ni-rich sulfide + theophrastite + hydroxyl Ni carbonates + $SO_2$.

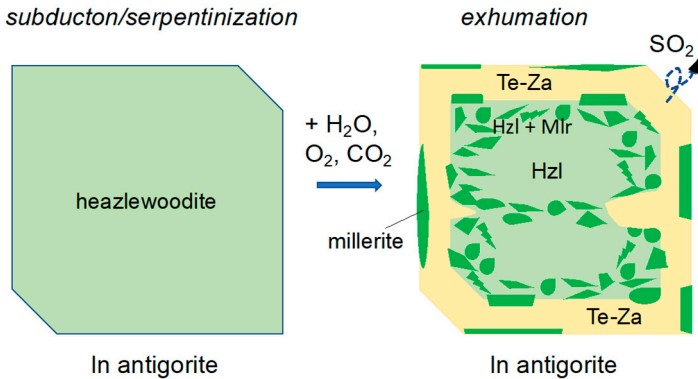

**Figure 8.** Schematic illustration of the process of desulfurization of heazlewoodite. The metamorphic heazlewoodite (Hzl, $Ni_3S_2$; pale green) has been decomposed to millerite (Mlr, NiS; green) and theophrastite [Te, $Ni(OH)_2$] and zaratite [Za, $Ni_3(CO_3)(OH)_4 \cdot 4H_2O$)] (dark yellow) during/after exhumation of the complex.

Pentlandite ([Ni, Fe]$_9S_8$; Ni > Fe), the most common nickel-rich sulfide in ultramafic rocks [65], may be decomposed to violarite ($FeNi_2S_4$), possibly with iron-rich sulfides (or oxides) during the formation of nickel hydroxide or hydroxyl carbonates [20].

The situation of nickel sulfide desulfurization is quite different if the serpentinization of olivine is simultaneously in progress. A highly reduced condition is achieved by production of hydrogen and magnetite ($Fe^{3+}$-bearing), mainly with awaruite [12,14,15] instead of nickel hydroxide or hydroxyl carbonates. That is, nickel is mainly stored not in hydroxide or hydroxyl carbonates, but in awaruite, because of the reduced condition with hydrogen. In a suboceanic peridotite from the Mid-Atlantic Ridge, millerite and polydymite are produced, instead of awaruite-pentlandite-heazlewoodite, in a steatized (talc-rich) part near gabbroic intrusions possibly via further desulfurization [14]. We then expect that the steatized abyssal serpentinite [14] possibly contains nickel hydroxide or hydroxyl carbonates. The formation of theophrastite-zaratite in the Fujiwara meta-dunite has been performed in a post-serpentinization stage, where little serpentinization of olivine has occurred and an oxidized environment was available.

## 6. Conclusions and Implications

The reaction above (Figure 8) is due to the transportation of the subduction complex to the oxidizing and hydrous environment at an early stage or during exhumation [66]. This represents one of the oxidation processes of the serpentinized peridotite exposed to the oxidizing environment near the surface. The subducted sulfur is in part stabilized in serpentinized peridotite as heazlewoodite,

and then released on its oxidation decomposition during exhumation. The oxidized sulfur liberated by this reaction has been transported and fixed as sulfates, such as anhydrite in the surrounding rocks in the Sanbagawa belt [67,68] (Figure 1). The anhydrite has been found from amphibolite, hornblendite and clinopyroxenite in a hydrated (retrogressively metamorphosed) part of the Iratsu body, overlying the Higashi-akaishi peridotite complex (Figure 1) [69,70]. Nickel-rich sulfides and arsenides similar to those in the Fujiwara meta-dunite have been also found from the Higashi-akaishi meta-peridotite [46], and we expect a similar desulfurization process in and around the Higashi-akaishi peridotite.

If we take the common occurrence of heazlewoodite in serpenetinized peridotites and chromitites [12,53,55,71] into account, we predict that nickel hydroxide and nickel hydroxyl carbonates are more common than we have thought in peridotites composing exhumed subduction complexes. Their formation occurs solely when the serpentinization of olivine does not occur, that is, most easily after serpentinization has been completed.

**Author Contributions:** Conceptualization, S.A. and S.I.; methodology, All authors; microprobe analysis, S.I., M.M. and N.A.; SEM analysis, S.I. and S.A.; Raman analysis, M.M. and T.M.; writing, All authors; editing, S.A. All authors have read and agreed to the published version of the manuscript.

**Funding:** This research was funded by Japan Mining Promotive Foundation to S.A. (Transportation and concentration processes of Cr and PGE by hydrothermal solutions) and JSPS KAKENHI Grant Numbers JP24540518 and JP16K1783400 to S.I.

**Acknowledgments:** R. Sawada kindly made thin sections used in this study. T. Nishiyama helped us in SEM analysis at Kumamoto University. Collaboration with H. Okamura in the field is highly appreciated. We thank the late H.M. Prichard for giving us information on the Shetland ophiolite. We appreciate critical comments by two anonymous reviewers, which were helpful in the revision of the manuscript.

**Conflicts of Interest:** The authors declare no conflict of interest.

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
