# Peer review of "Post-Serpentinization Formation of Theophrastite-Zaratite by Heazlewoodite Desulfurization: An Implication for Shallow Behavior of Sulfur in a Subduction Complex"

_minerals, doi:10.3390/min10090806_

Round 1

Reviewer 1 Report

This article is interesting and nicely written, in general.

- I have detected minor typos that should be corrected. See attachment.

- For the benefit of the readers, the paper should be restructured with a Methodology section (Raman and Mineral Chemistry), introducing a Result section with the results obtained from the analysis. See attachment.

- All along the paper, authors talk about antigorite and antigoritization, but I do not see any explanation of the real appearance of this serpentine phase and no other. For example, Endo et al. Talk about lizardite in this same context. Why antigorite??? I do not see the importance of being antigorite, instead of other serpentine polymorph, so either antigorite is changed for serpentine, or a explanation of how antigorite was detected (as differentiating serpentine polymorphs is somehow complicated) is included in the paper.

Ref.: Shunsuke Endo, Tomoyuki Mizukami, Simon R. Wallis, Akihiro Tamura, and Shoji Arai: Orthopyroxene-rich Rocks from the Sanbagawa Belt (SW Japan): Fluid–Rock Interaction in the Forearc Slab–Mantle Wedge Interface

Author Response

I have detected minor typos that should be corrected. See attachment.

Thank you very much. We have done the correction.

- For the benefit of the readers, the paper should be restructured with a Methodology section (Raman and Mineral Chemistry), introducing a Result section with the results obtained from the analysis. See attachment.

Thank you very much for this comment. But we do consider that the current form may be much more convenient for readers. We should make rather long statements for the methodology as compared with the result proper for both Raman spectroscopy and mineral chemistry. In addition, each methodology is related ONLY with the result. Then we consider the methodological statements are better to be left in Raman spectroscopy and Mineral chemistry sections as they are.

- All along the paper, authors talk about antigorite and antigoritization, but I do not see any explanation of the real appearance of this serpentine phase and no other. For example, Endo et al. Talk about lizardite in this same context. Why antigorite??? I do not see the importance of being antigorite, instead of other serpentine polymorph, so either antigorite is changed for serpentine, or a explanation of how antigorite was detected (as differentiating serpentine polymorphs is somehow complicated) is included in the paper.

Thanks for this important comment. We mean that the main serpentine species in the sample is antigorite. So, in this special case, “serpentinization” is “antigoritization”. And we do understand that the “antigoritization” was not exclusive to produce the theophrastite-zaratite formation. So, we replaced “antigoritization” with “serpentinization” as well as “antigoritized” with “serpentinized” in the new version.

Ref.: Shunsuke Endo, Tomoyuki Mizukami, Simon R. Wallis, Akihiro Tamura, and Shoji Arai: Orthopyroxene-rich Rocks from the Sanbagawa Belt (SW Japan): Fluid–Rock Interaction in the Forearc Slab–Mantle Wedge Interface

Thank you very much. S.A. was so careless to skip this important paper even though I am one of authors! We refer to this in the new version.

Reviewer 2 Report

Article Review

Journal:  Minerals

Title:  Post-Serpentinization Formation of Theophrasite-Zaratite by Heazlewoodite Desulfurization:  An Implication for Shallow Behavior of Sulfur in a Subduction Complex

Authors:  S. Arai, S. Ishimaru, M. Miura, N. Akizawa and T. Mizukami

This well written manuscript provides an in depth mineralogical assessment of regionally metamorphosed dunite from Fujiwara/Sanbagawa metamorphic belt and the desulfurization of heazlewoodite during exhumation.  One question I had was whether O2 could be the only component driving oxidation (i.e., producing an oxidizing environment); (Line 311/325.  Reactions)?  Even in highly reduced conditions, metastable H2O2 could drive oxidation or possibly other reaction pathways?  Overall, this is a well written manuscript appropriate for Minerals.

General Comments:

Line 25.  Please include the chemical formula for heazlewoodite.  It is mentioned as a nickel sulfide on Line 45, but please consider adding the formula here as well as it is the first mention of this mineral in the main text.

Lines 82 Unclear what is meant by ‘was suffered.’  Altered?

Line 89.  Would recommend change to:  Serpentinization, specifically antigoritization, has…

Line 105.  Name the Raman figure here in the Figure 3 caption versus saying ‘analysis below’ because the figure is a few pages away.

Line 152-169.  Double check that there are spaces between ## and unit of measurement.

Line 213.  No space between wt and % as consistent with other wt%

Line 216. Include measurement is in terms of wt% in Table 1.

Line 253.  SEM analyses are in terms of wt%, correct?

Lines 361.  Olivine is misspelled.

Line 258.  Does heazlewoodite commonly occur in serpentinized peridotites?  Please support this statement with references.

Line 361-362.  Could more explanation be provided as to why Ni hydroxide/carbonates occur solely when serpentinization of olivine does not occur?

Author Response

This well written manuscript provides an in depth mineralogical assessment of regionally metamorphosed dunite from Fujiwara/Sanbagawa metamorphic belt and the desulfurization of heazlewoodite during exhumation. One question I had was whether O could be the only component driving oxidation (i.e., producing an oxidizing environment); (Line 311/325. Reactions)? Even in highly reduced conditions, metastable H O could drive oxidation or possibly other reaction pathways? Overall, this is a well written manuscript appropriate for Minerals.

Thank you very much for the comment. But oxygen is the most probable, being easily available for the serpentinized peridotite during exhumation.

General Comments:

Line 25. Please include the chemical formula for heazlewoodite. It is mentioned as a nickel sulfide on Line 45, but please consider adding the formula here as well as it is the first mention of this mineral in the main text.

  1. We have done for millerite and heazlewoodite.

Lines 82 Unclear what is meant by ‘was suffered.’ Altered?

But we think this sufficiently means. We mean that the peridotite was regionally metamorphozed at the subduction zone.

Line 89. Would recommend change to: Serpentinization, specifically antigoritization, has…

Thank you very much. We agree and deleted “(antigoritization)”.

Line 105. Name the Raman figure here in the Figure 3 caption versus saying ‘analysis below’

because the figure is a few pages away.

Thanks for this comment. We would like to refer to the Raman spectroscopy issue after Petrography, so it is not a good idea to put the Raman figures here before Figure 4.

Line 152-169. Double check that there are spaces between ## and unit of measurement.

Thank you very much. We checked and corrected.

Line 213. No space between wt and % as consistent with other wt%

Corrected. Thanks.

Line 216. Include measurement is in terms of wt% in Table 1.

We have done. Thanks.